# Bioinformatics-Guided Identification of Ethyl Acetate Extract of Citri Reticulatae Pericarpium as a Functional Food Ingredient with Anti-Inflammatory Potential

**DOI:** 10.3390/molecules27175435

**Published:** 2022-08-25

**Authors:** Enyao Ma, Lu Jin, Chunguo Qian, Chong Feng, Zhimin Zhao, Hongru Tian, Depo Yang

**Affiliations:** 1Guangdong Hanchao Traditional Chinese Medicine Technology Co., Ltd., Guangzhou 510163, China; 2School of Pharmaceutical Sciences, Sun Yat-sen University, Guangzhou 510006, China; 3Guangzhou Caizhilin Pharmaceutical Co., Ltd., Guangzhou 510360, China

**Keywords:** Citri Reticulatae Pericarpium, anti-inflammatory functional food, network pharmacology analysis, molecular docking, HPLC/MS/MS-based chemical profiling

## Abstract

Citri Reticulatae Pericarpium (CRP) is one of the most commonly used food supplements and folk medicines worldwide, and possesses cardiovascular, digestive, and respiratory protective effects partially through its antioxidant and anti-inflammatory functions. The unique aromatic flavor and mild side effects make CRP a promising candidate for the development of anti-inflammatory functional food. However, recent studies show that the crude alcoholic extract and some isolated compounds of CRP show compromised anti-inflammatory activity, which became the main factor hindering its further development. To identify the bioactive compounds with anti-inflammatory potential, and improve the anti-inflammatory effects of the extract, a bioinformatics-guided extraction protocol was employed in this study. The potential bioactive candidates were identified by combing network pharmacology analysis, molecular docking, principal components analysis, k-means clustering, and in vitro testing of reference compounds. Our results demonstrated that 66 compounds in CRP could be grouped into four clusters according to their docking score profile against 24 receptors, while the cluster containing flavonoids and phenols might possess a more promising anti-inflammatory function. In addition, in vitro anti-inflammatory tests of the seven reference compounds demonstrated that hesperitin, naringenin, and gardenin B, which were grouped into a cluster containing flavonoids and phenols, significantly decreased LPS-induced NO, TNF-α, and IL-6 production of macrophages. While the compounds outside of that cluster, such as neohesperidin, naringin, hesperidin, and sinensetin showed little effect on alleviating LPS-induced NO and proinflammatory cytokine production. Based on the chemical properties of selected compounds, ethyl acetate (EtOAc) was selected as the solvent for extraction, because of its promising solubility of flavonoids and phenols. Furthermore, the ethanol alcoholic extract was used as a reference. The chemical profiling of EtOAc and crude alcoholic extract by HPLC/MS/MS also demonstrated the decreased abundance of flavonoid glycosides in EtOAc extract but increased abundance of phenols, phenolic acid, and aglycones. In accordance with the prediction, the EtOAc extract of CRP, but not the crude alcoholic extract, significantly decreased the NO, IL-6, and TNF-α production. Taken together, the results suggested selective extraction of phenols and flavonoids rich extract was able to increase the anti-inflammatory potential of CRP partially because of the synergistic effects between flavonoids, phenols, and enriched polymethoxyflavones. Our study might pave the road for the development of ethyl acetate extract of CRP as a novel functional food with anti-inflammatory function.

## 1. Introduction

Inflammation, especially chronic inflammation, is commonly viewed as the driving factor in the initiation and progression of various diseases, such as cancer, autoimmune diseases, cardiac vascular diseases, neurodegenerative diseases, and diabetes [1,2], which is characterized by infiltration and the continued recruitment of mononuclear leukocytes and tissue injury [3]. Within the infiltrated cells, the macrophage plays an essential role in the pathogenesis of chronic inflammation by secreting a panel of mediators and cytokines including interleukin-1, interleukin-6, interleukin-12, TNF-α, and inducible nitric oxide synthase (iNOS) after activation [4]. An increasing body of evidence demonstrates that switching the “proinflammatory” macrophages to the “anti-inflammatory” subtype may serve as a potential therapeutic way of alleviating tissue damage or syndromes associated with chronic inflammation [5], which makes macrophages continue to be an attractive therapeutic target for the prevention or treatment of chronic inflammation [6].

A variety of effective anti-inflammatory agents are available or under development nowadays, including nonsteroidal anti-inflammatories, antibodies against proinflammatory cytokine, small molecules that block the activity of kinases, histone deacetylase inhibitors, PPAR agonists, and even small RNAs [7]. However, long-term usage of anti-inflammatory drugs may cause gastrointestinal side effects and increased risk of cardiac vascular diseases and progressive multifocal leukoencephalopathy [8]. Thus, long-term safety is still one of the problems that many of the currently available therapies of anti-inflammatory drugs need to face. Recently, food supplements or functional foods emerged as an alternative treatment for inflammations, which can alleviate the progression of inflammation by modulating inflammatory cytokines [9]. Unlike the available anti-inflammatory agents, functional foods commonly show relatively mild adverse effects. In addition, functional foods or food supplements may benefit patients with chronic inflammation by regulating inflammation-related signaling pathways [10].

Citri Reticulatae Pericarpium (CRP) is a commonly used food complement and functional food and folk medicine [11] which possesses promising anti-oxidative [12], anti-viral, anti-bacterial [13], anti-tumor [14], and immunoregulatory functions [15]. The unique aromatic flavor and mild side effects make CRP a promising candidate for the development of anti-inflammatory functional food. However, the development of CRP as a functional food involves at least two problems: identifying potential bioactive candidates and selecting organic solvents to enrich them. Furthermore, the extraction method was demonstrated to play a role in affecting the extracting efficacy. For example, two commonly used methods for the extraction of phenols and flavonoids from medicinal plants, reflux and ultrasonic extraction. share similar extraction yields and recoveries, however, ultrasonic extraction shows increased extraction efficacy and lower extraction duration [16]. Although the majority of studies confirm that flavonoids and their glycosides play an essential role in their anti-inflammatory function, the anti-inflammatory activity of flavonoids, especially the polymethoxyflavones and flavonoid glycosides, was still in debate, and some studies showed that the flavonoids and their glycosides were able to inhibit Th2 response [17,18,19], indicating their potential effects on inhibiting macrophage M2 polarization. On the other hand, CRP contains various compounds with multi-target capabilities, which may participate in the anti-inflammatory and immunostimulating receptors or signaling pathways. The dual pro- and anti-inflammatory regulatory features of flavonoids and their glycosides will make the identification of bioactive candidates and the subsequent organic solvation selection more difficult.

To solve these problems, a systems pharmacology-based compound identification protocol was employed in the current study (as shown in Figure 1). Systems pharmacology, which is based on the theory of systems biology, integrating multi-disciplinary technologies such as multi-directional pharmacology, bioinformatics, and computer science, provides us with a useful tool with which to analyze the complex function of TCM by constructing multi-level “function-gene-target-drug” networks [20,21]. In addition, with the help of molecular docking, the mode of interaction between selected receptors and drug molecules and the binding free energy can be calculated [22]. The combination of network pharmacology and molecular docking might provide us with an alternative way to identify the bioactive candidates from the perspective of potential targets or the biological function.

In the current study, network pharmacology and molecular docking were employed to select the potential target and predict the binding free energy between compounds in CRP, respectively. Based on the analysis of the patterns beneath the docking score profile, the chemical properties of selected compounds were calculated, and the extracting solvent was selected. Finally, the chemical profile and anti-inflammatory activity of the selected extract and the crude ethanol alcoholic extract were compared with HPLC/MS/MS and in vitro assays, respectively.

## 2. Results and Discussion

### 2.1. Network Pharmacological Analysis Predicted TLR4, PI3K/Akt, JAK/STATs, MAPK, and mTOR Signaling Pathways Were Closely Related to the Anti-Inflammatory Function of Citri Reticulatae Pericarpium Extract

The compounds isolated from Citri Reticulatae Pericarpium and their potential targets were retrieved from the TCMSP database [23] (https://tcmsp-e.com/tcmsp.php (accessed on 7 May 2022)). According to the chemical scaffold, 66 compounds were subdivided into eight different categories, such as flavonoids and flavones (4), polymethoxyflavones (6), flavonoids glycoside (4), phenols and phenolic acid (11), sesquiterpenoids (5), monoterpenoids (12), fatty acids (18), and others (6). The predicted targets retrieved from the TCMSP database are listed in Appendix A. Similar to other natural products, the majority of compounds showed compromised drug-likeness [24]. All compounds possessed promising oral bioavailabilities (>0.2), indicating that the compounds can be ingested orally (as shown in Figure 2A). Considering that macrophage polarization plays an essential role in regulating the phenotype and function of macrophages, the relationship between the predicted targets of bioactive compounds and the receptors that participated in macrophage polarization (listed in Appendix A) was analyzed with a Venn diagram. As shown in Figure 2B, eighty-six receptors were within the intersection of two parts, indicating that the receptors were involved in the immune-regulatory function of Citri Reticulatae Pericarpium. To better understand the underlying mechanism of the anti-inflammatory function of the bioactive ingredients of CRP, a network pharmacology model was constructed. The initial network pharmacology model, which comprised 11,626 nodes and 60,495 interactions, was filtered by the degree of nodes (greater than 20) and betweenness (greater than 0.001), and a core network comprising 335 nodes and 12,304 interactions was constructed. The biological function of selected receptors was predicted with enrichment analysis with GlueGO, and the results showed that selected receptors participated in the regulation of cytokine production, Th1–Th2 differentiation, regulation of microtube dimerization, and response to external stimulation (Figure 2C). The enrichment analysis also highlighted that TLR4, PI3K/Akt, JAK/STATs, MAPK, and mTOR signaling pathways, nitric oxide biosynthetic process, and autophagy (Figure 2D) might contribute to the anti-inflammatory function of CRP.

### 2.2. The Proposed Bioactive Candidate Identification Model Separated the Identified Compounds into Four Clusters According to Their Docking Score Profiles

The network pharmacology analysis and enrichment analysis not only predicted potential pathways which might be involved in the anti-inflammatory function of CRP, but highlighted 24 receptors according to their degree and betweenness, including ITGAM, TLR4, RAC1, CSF1R, SYK, TYK2, PI3K, AKT1, JAK1, JAK2, JAK3, STAT1, STAT3, STAT5, cSRC, mTOR, PDE3A, JNK, MAPK14, PPARγ, IKK, NOS2, COX2, and Tubulin. Although all of these receptors participated in the inflammation, their biological functions differed. For example, the activation of TLR4, Rac1, Src, JNK, and STAT1 was demonstrated to promote the transcription and translation of pro-inflammatory cytokines [25,26], while the activation of PPARγ, STAT3, MAPK14, and mTOR was believed to exert anti-inflammatory activity by polarizing macrophages to M2-like direction [27,28]. As a result, combinatorial effects of the bioactive ingredients which acted on the “Network targets” of macrophage polarization might finally determine its phenotype [29].

Accordingly, the 24 receptors involved in pro- and anti-inflammatory functions were employed. Although including receptors participating in both pro- and anti-inflammatory processes would increase the complexity of interpreting results obtained from molecular docking, a systematic evaluation of the balance between pro- and anti-inflammatory potential might increase the discrimination ability and accuracy of our candidate identification model. Considering that the majority of natural products, such as flavonoids, phenols, or lignans are believed to function as ATP or another co-factor competitive inhibitor [30], the ATP binding pocket or co-factor binding pocket of the above-mentioned receptors was defined as an active site.

As shown in Figure 3A, the flavonoids, polymethoxyflavones, and phenols showed relatively high predicted binding affinity to the above-mentioned targets, which was consistent with the literature data in that the flavonoids, flavanone glycosides, and polymethoxyflavones of Citri Reticulatae Pericarpium are believed to be the bioactive ingredients in CRP with anti-inflammatory function [31]. Furthermore, we noticed that some monoterpenoids or sesquiterpenoids showed moderate predicted binding affinity to the receptors. Unlike flavonoids or phenols, the fatty acid showed the lowest predicted binding affinity towards all receptors.

To extract the underlying patterns of the docking score profiles and find the similarity between compounds, the docking profiles were subjected to principal component analysis. As shown in Figure 3B, the first three components, PC1, PC2, and PC3, captured 89.8% of total variations, indicating the reliability of the model. The subsequent k-means clustering grouped the compounds into four clusters (Appendix A). As shown in the 2-D projection (Figure 3C), the majority of the fatty acids were grouped into cluster IV, at the same time, all the monoterpenoids were grouped into cluster III. The flavonoids, polymethoxyflavones, and their glycosides, which were believed to be the bioactive ingredients in CRP, were subdivided into three clusters. All the flavonoids, such as hesperitin, naringenin, (R)-hesperetin, and quercetin, as well as the phenols, fell into cluster I, while the glycosides were grouped into cluster II.

Interestingly, the polymethoxyflavones were subdivided into two different clusters according to the docking score profile: gardenin B, tangeretin, and nobiletin were in Cluster I, while the remaining polymethoxyflavones were in Cluster III. This interesting finding led us to question whether the observation was of biological relevance. Unexpectedly, the anti-inflammatory potential of polymethoxyflavones also varied remarkably according to the literature data. For example, Ho, et al. found that only nobiletin displayed a promising inhibitory effect on LPS-induced proinflammatory cytokines or NO secretion, while other polymethoxyflavones and flavone glycosides showed very weak anti-inflammatory activity [32]. Similarly, Tominari, et.al found that tangeretin and nobiletin were able to alleviate LPS-induced proinflammatory cytokine production in Raw264.7 cells [33]. On the contrary, 3,5,6,7,8,3′,4′-Heptamethoxyflavone, another polymethoxyflavone found in CRP, was able to inhibit IL-4 production and might limit macrophage M2 polarization [34]. The results from our model showed a clear separation between gardenin B, tangeretin, nobiletin, and heptamethoxyflavone or sinensetin, which was in accordance with the literature data. Furthermore, the literature data also suggested that the compounds in cluster I might possess more potent anti-inflammatory activity than other clusters. For example, Ren, et al. demonstrated that hesperetin dose-dependently reduced TNF-α, IL-6, and IL-β secretions in LPS-induced murine macrophages [35]; naringenin alleviated LPS-induced IL-1β, IL-6, IL-8, and TNF-α released by macrophages [36,37]. Hesperidin, the glycoside of hesperitin, showed less potent COX inhibitory activity and anti-inflammatory activity than its aglycone [38,39].

### 2.3. In Vitro Anti-Inflammatory Evaluation of Reference Compounds Confirmed That the Cluster Comprising Flavonoids, Phenols, and Phenolic Acid Might Possess Potential Anti-Inflammatory Functions

To further validate the reliability of our bioactive candidate selection model, the in vitro anti-inflammatory function of seven representative compounds was tested. As shown in Figure 4, hesperitin, naringenin, and gardenin B, which belonged to cluster I, could inhibit LPS-induced NO production, IL-6, and TNF-α release of Raw 264.7 macrophages in a dose-dependent way, indicating that they possessed potential anti-inflammatory function. However, neohesperidin, naringin, and hesperidin, which belonged to cluster II, and sinensetin (which belonged to cluster III), showed little effect on LPS-induced NO, IL-6, and TNF-α production. The results from in vitro testing with reference compounds further confirmed that the cluster comprising flavonoids, phenols, and phenolic acid might possess potential anti-inflammatory functions.

On the other hand, a detailed analysis of the interactions between seven reference compounds and six representative receptors was performed. The chosen receptors comprised kinases (JNK, AKT1, and mTOR), nuclear receptors (PPARγ), the signal transducer and activator of transcription family member (STAT1), and enzymes that participated in the inflammatory process (COX2), which possessed not only differing molecular functions but also varied in the size and geometry of active sites. For example, the nuclear receptors (such as PPARγ) and COX2 possessed a closed and hydrophobic pocket for the ligand, while the kinases catalytic domain showed a semi-closed pocket for the ligand. On the contrary, the STATs exhibited a shallow ligand binding pocket close to the surface. The results from ligand–receptor interaction analysis demonstrated that the driving force in stabilizing the predicted binding pose differed remarkably among the reference compounds.

The polar interactions between the polar group or the saccharide motif of flavonoid glycosides were demonstrated to be the main driving force in stabilizing their binding pose within selected receptors. As shown in Figure 5 and Figure 6, a variety of hydrogen bonds or electrostatic interactions between the saccharide motif and polar residues (such as Glu109, Asn144, See34, Ser155 in JNK, Thr82, Asp292, Thr211, Tyr272, Ser205 in AKT1, or Thr2245, Asp2357, Tyr2225, Asp2195 in mTOR, Ser606, Asp609, Asp632, Lys584 in STAT1, and Asp295, Asn343m Ala342 in PPARγ). Furthermore, the numbers of hydrophilic and hydrophobic interactions between flavonoid glycosides and kinases were similar, indicating that the flavonoids may affect the activity of kinases with anti-inflammatory activity. The antagonistic effects due to their lower selectivity may affect their overall anti-inflammatory potential. In addition, we found that the flavonoid glycosides could not fit into the ligand binding site of PPARγ and COX-2, which may further decrease their anti-inflammatory effects.

The hydrophobic interactions played an essential role in stabilizing the predicted binding pose of flavonoids and polymethoxyflavones. The flavonoids and polymethoxylflavones penetrated the inner site of the semi-open binding pocket of kinases, as well as the closed binding pocket of COX-2 and PPARγ, and stabilized mainly through the hydrophobic interactions between surrounding hydrophobic residues (such as M108. V40, I32, M110, and L110 in JNK; Y272, V271, V270, W80, and L264 in AKT1; W2239, L2185, L2192, I2237, I2356, and M2345 in mTOR; Tyr473, Leu469, Leu330, Phe327, Val270, Ile363, and Met453 in PPARγ; Ile517, Phe518, Val349, Leu531, Val116, Val523, and Tyr355 in COX-2). In addition, we found that the flavonoids fell into the open and shallow pocket of STAT1 and stabilized mainly through hydrophobic interactions with surrounding hydrophobic residues, such as Phe628, Val631, Tyr645, and Trp616, as well as the polar interactions between ligands with Ser606, Glu609, Asp638, and Asp632. The increased contribution of hydrophobic interaction in flavonoids and polymethoxylflavones may not only compensate for the decreased stability caused by decreased polar interactions (presented in flavonoid glycosides) but also increase the stability and selectivity of ligands towards the receptors with more hydrophobic binding pockets (such as JNK). The increased binding affinity towards kinases and enzymes that participated in the pro-inflammatory process may provide a plausible explanation for the observed increased anti-inflammatory capacity of hesperitin and naringenin.

To selectively enrich the compounds with anti-inflammatory functions, the chemical and physical properties, such as oral bioavailability, molecular weight, logP, and caco2 cell permeability were analyzed (Appendix A). Because the candidates comprised flavonoids, flavones, phenols, and phenolic acid, which showed promising solubility in ethyl acetate, the EtOAc was finally selected as the extraction solvent.

### 2.4. Chemical Profiling and In Vitro Anti-Inflammatory Test Demonstrated That Flavonoids and Phenols Were Enriched in the EtOAc Extract, Which Might Contribute to its Increased Anti-Inflammatory Potential

The chemical profiles of crude ethanol extract and the EtOAc extract were analyzed by HPLC/MS/MS, and the retention time, accurate mass, mass defect, formation of negative/positive ion, and fragmentation patterns, as well as literature matching, allowed putative identification, i.e., identification without standards, of compounds within two extracts. For example, the secondary metabolites with some degree of hydrophilicity, such as flavonoids and glycosides, were eluted after 1–5 min, while the secondary metabolites, such as flavonoids, polymethoxyflavones, possessed an increased degree of hydrophobicity and eluted in the middle part of the elution gradient after 7–25 min. As expected, the EtOAc extract showed an increased abundance of elution after 10–25 min (as shown in Figure 7A). The results from HPLC/MS/MS analysis identified twenty-five compounds by comparing the retention time, elementary composition, and fragmentation with the literature (Figure 7B and Appendix A). In agreement with our prediction, the abundances of caffeic acid, ferulic acid, hesperitin, and gardenin B all increased in the EtOAc extract, while the abundances of glycosides (such as vicenin, stellarin, and naringin) decreased in the EtOAc extract. In addition, we found that the abundance of other lipophilic compounds, such as isosinensetin, sinensetin, and xanthomicrol, increased in EtOAc extract.

Next, the anti-inflammatory activities of EtOAc extract and the crude alcoholic extract were compared. As shown in Figure 7C, the EtOAc extract possessed more potent anti-inflammatory activity than crude alcoholic extract, and 50 μg/mL EtOAc extract was able to significantly decrease LPS-induced NO, IL-6, and TNF-α secretion, indicating that the EtOAc extract might possess more potent anti-inflammatory activity than crude alcoholic extract in vitro. Some studies consider that flavonoids and polyphenols possess better bioavailability because of their increased lipophilicity and better bioactivity [40], however, the results from our study showed that the EtOAc and crude alcoholic extract possessed very similar chemical profiles, where the abundance of hydrophobic compounds, such as hesperitin, nobiletin, and sinensetin, was also detected in the crude alcoholic extract, indicating the difference in anti-inflammatory potential might result from the presence of beneficial “synergistic” interactions and “antagonism” interactions.

It was widely accepted that traditional Chinese medicine could maximize drug efficacy and minimize adverse effects via the characteristics of “multiple components, multiple targets, and multiple pathways” [41]. The complex interactions between varied compounds might not only enhance the bioavailability of active components, but also promote therapeutic effects, and/or reduce toxicity by the synergistic or additive effects [42]. In the current study, we found the increased anti-inflammatory potential of EtOAc extract of CRP might partially be due to the synergistic effects between flavonoids, phenols, and polymethoxyflavones. The HPLC/MS/MS-based chemical profiling demonstrated that the polymethoxyflavones were still the main ingredients of EtOAc, as well as the EtOH extract. According to our prediction, the polymethoxyflavones (such as gardenin B and nobiletin) showed relatively high binding affinities with TLR4, JNK, and COX2, however, their binding affinity with STAT1 and IKK was not quite as promising as flavonoids or phenols. In the EtOAc extract, the potential synergistic effects between flavonoids and polymethoxyflavones might increase its overall anti-inflammatory efficacy by inhibiting multiple signaling pathways, such as TLR4, JAK/STAT1, JNK, and their downstream enzymes (such as COX2 and iNOS). However, an increased abundance of flavonoid glycosides in the alcoholic crude extract of CRP was observed. The docking score profile demonstrated that the flavonoid glycosides exhibited higher binding affinities with mTOR and Akt than their aglycones. It was reasonable to predict that the increased abundance of glucosides in the crude alcoholic extract might mask the anti-inflammatory activities of flavonoids or polymethoxyflavones by functioning as competitive inhibitors to anti-inflammatory targets and decreasing their anti-inflammatory potential because of the antagonistic effects.

Additionally, the abundance of nobiletin and tangeretin, two major polymethoxyflavones ingredients, did not change significantly within the crude alcoholic extract and EtOAc extract of CRP. Considering they showed compromised anti-inflammatory activities, it is still necessary to decrease their concentration by choosing another organic solvent, using a mixture of different organic solvents, or converting the flavonoid glycosides or polymethoxyflavones to bioactive derivates by chemical reactions.

## 3. Materials and Methods

### 3.1. Chemical Reagents and Extracts

Hesperitin was purchased from ACMEC, naringenin was purchased from Macklin, naringin was purchased from GuoHua Reagent Co., Ltd. (http://www.gh-reagent.com/, (accessed on 9 June 2022), Shanghai, China), hesperidin was purchased from J&K Chemical (Beijing, China), while neohesperidin, gardenin B, and sinensetin were purchased from Sichuan Weikeqi Biological Technology Co., Ltd. (Chengdu, China). Lipopolysaccharides (LPSs) from *Escherichia coli* O111:B4 was purchased from Sigma-Aldrich Chemical Co. (St., Louis, MO, USA). Dulbecco’s modified eagle’s medium (DMEM), fetal bovine serum (FBS), penicillin, and streptomycin were purchased from Gibco Life Technologies (Grand Island, NY, USA). The Griess assay kit was obtained from Beyotime Biotechnology (Shanghai, China). Mouse ELISA kits to detect TNF-α, IL-6, and IL-10 were purchased from Neobioscience Technology Co., Ltd. (Shenzhen, China).

The Citri Reticulatae Pericarpium sample was purchased from Changye Co., Ltd. (Guangzhou, China), and the crude alcoholic extract or EtOAc extract was prepared as follows: CPR samples were crushed into 200 mesh size and extracted with 100 mL 70% ethanol alcohol or extracted with 100 mL EtOAC for 30 min (300 w, 40 °C) under ultrasonic. Ethanol alcohol and EtOAc of analytical grade were supplied from DaMao Reagent (Tianjin, China). The extraction process was repeated twice, and the combined supernatant was filtered, concentrated in an evaporator under reduced pressure, and further dried to solid extract under a vacuum. The final extract was dissolved in methanol or DMSO and filtered with the 0.22 μm membrane filter for LC/MS/MS analysis or in vitro test.

### 3.2. Network Pharmacology Analysis

The bioactive ingredients and their potential targets of Citri Reticulatae Pericarpium were retrieved from the TCM systems pharmacology database and analysis platform (TCMSP) database [23] (https://tcmsp-e.com/tcmsp.php (accessed on 7 May 2022)) and used to construct the compound–target network. The target names were converted to gene symbols by searching the DrugBank database. A total of 135 proteins that were highly relevant to macrophage activation and polarization [43,44] were collected and used for the network construction. Finally, a compound–protein Interactions network was constructed by connecting identified nodes with protein–protein interaction information from STRING, HPRD, HIPPE, and Reactome databases as described [45]. The initial network was consolidated by the NetworkAnalyzer application of Cytoscape 3.7.1. The enrichment analysis and biological annotations were performed by ClueGo [46]. The network was visualized by Circos [47].

### 3.3. Molecular Docking and Analysis

Molecular docking was employed to predict the binding free energy between identified compounds with AKT1(PDBID: 4EJN), PI3K(PDBID: 3L54), JNK(PDBID: 3ELJ), PDE3A(PDBID: 7L29), NOS2(PDBID: 4NOS), NOS1(PDBID: 6CIC), SYK(PDBID: 1XBC), STAT1(PDBID: 1BF5), STAT3(PDBID: 1BG1), STAT5(PDBID: 1Y1U), cSRC(PDBID: 2BDF), PPARG(PDBID: 3GZ9), TLR4(PDBID: 3VQ1), MAPK14(PDBID: 1A9U), MTOR(PDBID: 4JT6), ITGAM(PDBID: 1IDO), JAK1(PDBID: 3EYG), JAK2(PDBID: 4D0X), JAK3(PDBID: 3LXK), IKK PDBID: (4KIK), RAC1(PDBID: 5N6O), TYK2(PDBID: 4GIH), CSF1R(PDBID: 3LCO), Tubulin(PDBID: 1SA0), COX2(PDBID: 6COX) according to literature [48,49,50,51,52,53,54,55,56,57,58,59,60,61,62,63]. The 2-D structures of the small molecules were retrieved from the PubChem database and converted to 3-D structures by Open Babel. The 3-D structures of protein and small molecules were parameterized by AutoDockTools, and Autodock 4.2 was used for molecular docking [64]. In the docking process, the Lamarckian genetic algorithm (LGA) was used for conformational search, and 50 conformations were generated for each protein. The top-scoring pose (conformation with lowest binding free energy) was collected for each ligand–receptor pair and used for constructing the binding free energy profile. The docking protocol was validated by redocking of the crystallographic ligand, and the RMSD of re-docked ligand compared to co-crystal ligand was listed in Appendix A. The binding free energy profiles of identified compounds and receptors were subjected to principal component analysis (PCA) and k-means clustering. The PCA model and clusters were constructed using the scikit-learn package in Python 3.9. The docking score profile and results from PCA were visualized by the Seaborn package. The predicted binding pose of the protein–ligand complex was visualized with PyMOL (The PyMOL Molecular Graphics System, Version 1.2 Schrödinger, LLC, New York, NY, USA).

### 3.4. Cell Culture and Treatment

The murine macrophage cell (RAW 264.7) was purchased from the Cell Bank of Shanghai Institute of Biochemistry and Cell Biology (Chinese Academy of Sciences, Shanghai, China), and maintained in DMEM high-glucose medium supplemented with 10% heat-inactivated FBS, 100 U/mL penicillin, and 100 μg/mL streptomycin at 37 °C in a humidified atmosphere of 5% CO_2_ according to the recommendation of ATCC. The cells were subcultured every 2–3 days, and the initial density was 300,000 cells/cm^2^.

### 3.5. Analysis of Macrophage Cytokines and NO Production

The effects of compounds or extracts on NO and pro-inflammatory cytokine secretion of macrophages were determined by Griess reaction and ELISA, respectively. RAW 264.7 cells (30,000 cells/well) were seeded in 96-well plates and incubated under 37 °C and 5% CO_2_ for 24 h. After removing the old medium, the macrophages were treated with compounds or extract solution (dissolved in fresh medium) with indicated concentration for 24 h in the presence of 100 ng/mL LPS. The 0.1% DMSO was used as the vehicle control. After treatment, the NO, IL-6, and TNF-α secreted by macrophage RAW 264.7 cells were determined by Griess reaction, or ELISA kit according to the manufacturer’s instructions, respectively. Cell viability was routinely tested by MTT assay, and the selected compounds, as well as the extracts, did not affect the viability of macrophages up to 100 μM and 50 μg/mL, respectively (data not shown).

### 3.6. HPLC/MS/MS-Based Chemical Characterization of Crude Alcoholic Extract of Citri Reticulatae Pericarpium and Its EtOAc Extract

Chemical characterizations of Citri Reticulatae Pericarpium extract and its EtOAc extract were carried out by high-performance liquid chromatography coupled with mass spectrometry (Waters SYNAPTG2-Si, Waters, Milford, MA, USA). For separation, a Waters ACQUITY UPLC BEH C18 Colum, 1.7 μm, 50 × 2.1 mm column was employed. The flow rate was set to 300 μL/min and the injection volume was 5 μL. The mobile phase consisted of eluent A, MilliQ water, and eluent B, acetonitrile, and both of them were acidified with 0.1% formic acid. The elution gradient was set to 0–8 min, 10–30% of B; 8–22 min 30–45% of B; 22–27 min, 45–95% of B; 27–31 min, 95% of B, 31–34 min, 10% of B. The MS analysis was performed in the positive and negative electrospray ionization mode, at a resolving power of 40,000 over a full scan range of *m*/*z* 50–1200. The compounds were identified by molecular weight and fragments.

### 3.7. Statistical Analysis

One-way ANOVA with Tukey post hoc test was used for the statistical analysis. Significant difference was considered when * *p* < 0.05, ** *p* < 0.005, *** *p* < 0.001. The analyses were performed using GraphPad Prism^®^ version 7.0 (GraphPad Software, Sab, San Diego, CA, USA).

## 4. Conclusions

In summary, combining network pharmacology analysis, molecular docking, principal component analysis, and k-means clustering, the flavonoids, phenols, and certain polymethxyflavones, but not flavonoid glycosides, were predicted to possess the potential for anti-inflammatory activity. Furthermore, our model also predicted that the EtOAc extract of CRP, which contained an increased abundance of bioactive ingredients and decreased abundance of flavonoid glycosides, might exhibit more promising anti-inflammatory potential than crude alcoholic extract. The hypothesis was proved by HPLC/MS/MS-based chemical profiling and in vitro tests. Unlike previous studies that considered the increased anti-inflammatory potential of flavonoids and polyphenols merely because of their increased lipophilicity and better bioactivity, the results of our study indicated the synergistic effects between flavonoids, phenols, and polymethoxyflavones might contribute to increased anti-inflammatory efficacy. As a result, the EtOAc extract of CRP held the potential to act as a natural immunomodulatory ingredient or functional food.

## Figures and Tables

**Figure 1 molecules-27-05435-f001:**
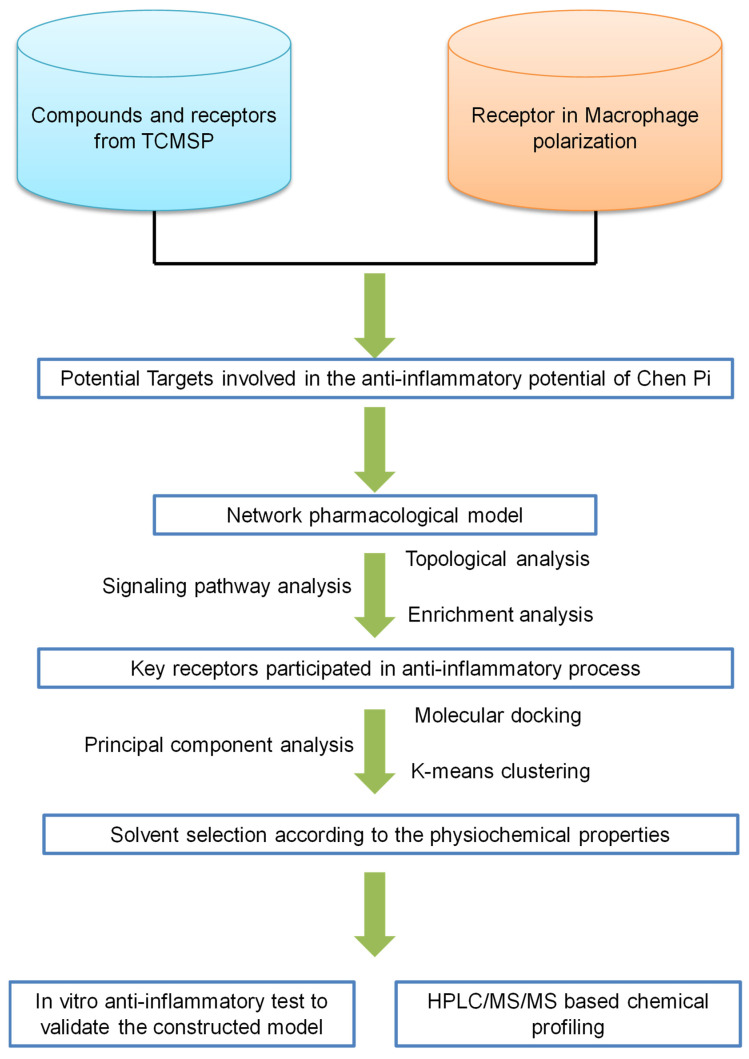
Schematic diagram of the protocol used in this study.

**Figure 2 molecules-27-05435-f002:**
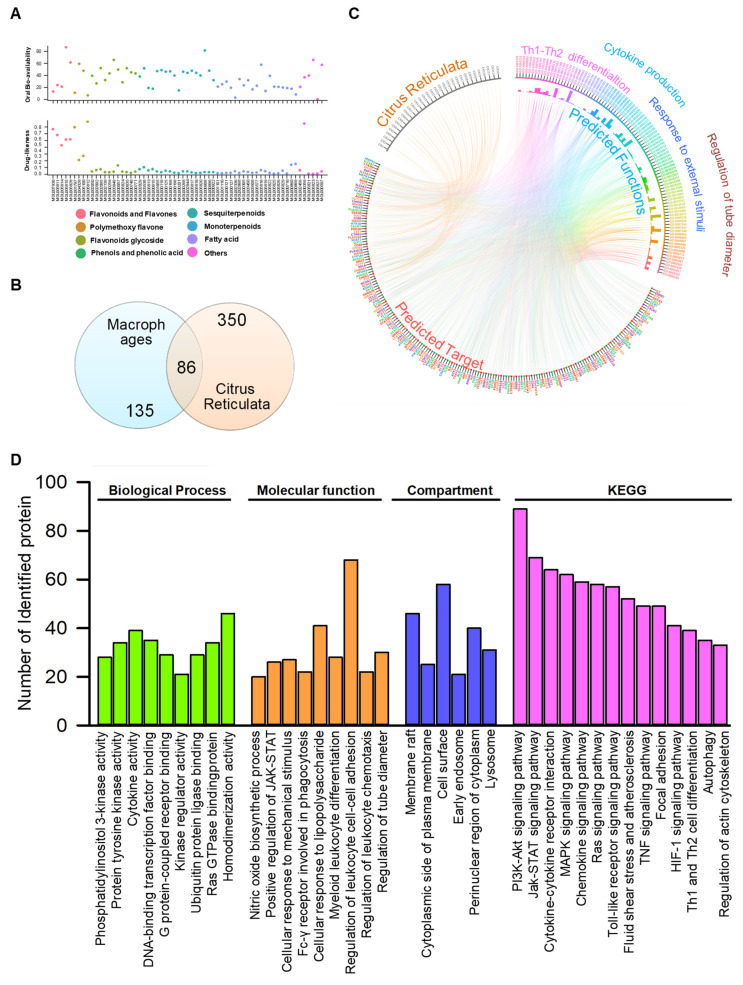
Network pharmacological analysis predicted that TLR4, JAK/STATs, MAPK, and mTOR signaling pathways were closely related to the anti-inflammatory function of Citri Reticulatae Pericarpium (CRP) extract. (**A**) Oral bioavailability and drug-likeness of identified compounds from CRP. The oral bioavailability and drug-likeness of identified compounds were retrieved from the TCMSP database and visualized as a scatter plot. The relationship between predicted targets and the proteins that participated in the regulation of macrophage polarization was plotted in (**B**). (**C**) Network pharmacology-based analysis of identified compounds in CRP and its related biological functions. (**D**) The biological functions of identified compounds in CRP arranged into networks by the STRING database using gene ontology and KEGG datasets.

**Figure 3 molecules-27-05435-f003:**
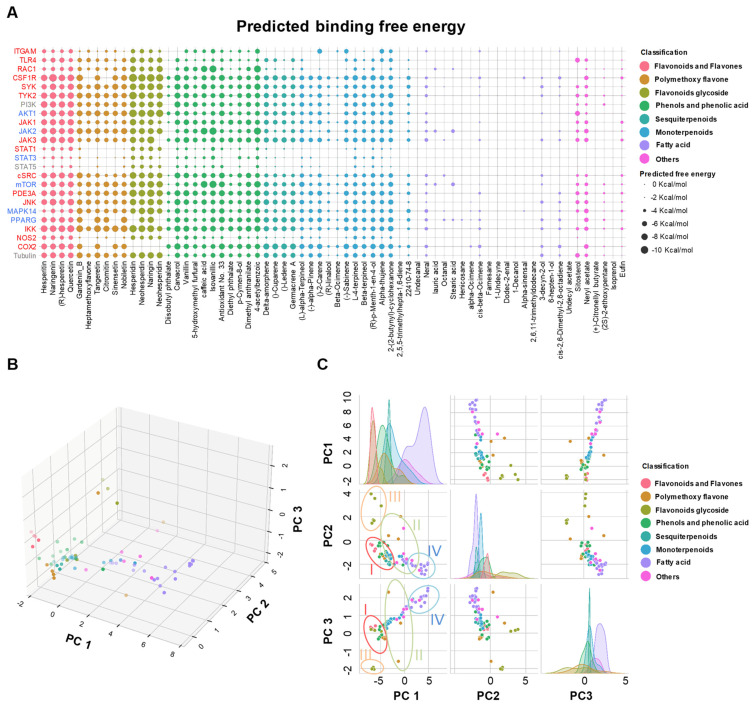
Principal component analysis separated the identified compounds into 4 clusters according to the docking score profile against selected receptors. (**A**). Docking score of identified compounds with selected receptors. The receptor that mainly participated in M1 polarization was colored red, while the receptor that mainly mediated M2 polarization was colored blue. The compounds were colored according to their categories. (**B**). The results from PCA and the 2-D projection of the 3-D model were plotted in (**C**). The compounds in (**B**,**C**) were colored according to their categories.

**Figure 4 molecules-27-05435-f004:**
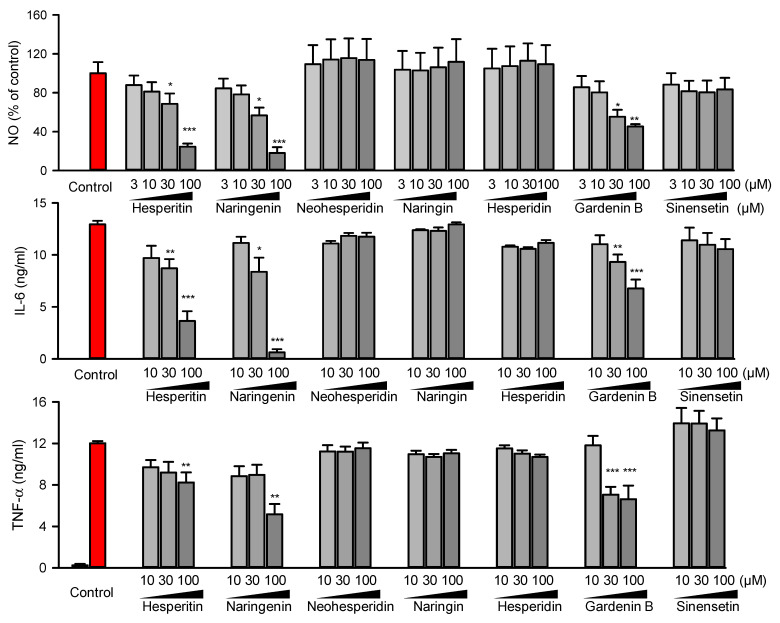
In vitro experiments and docking analysis identified that phenols, phenolic acid, and flavonoids possess anti-inflammatory activity. Hesperitin, naringenin, and gardenin B reversed LPS-induced TNF-α, IL-6, and NO release in a dose-dependent way. Raw264.7 macrophages was treated with vehicle control or compounds with indicated concentration for 24h, the cytokines was measured with ELISA, and normalized to the positive control (colored with red in the bar chart). The pooled data were visualized with mean ± SEM of at least three independent experiments. The one-way ANOVA and Tukey post hoc tests were employed for statistical analysis. *: *p* < 0.05, **: *p* < 0.01, ***: *p* < 0.001.

**Figure 5 molecules-27-05435-f005:**
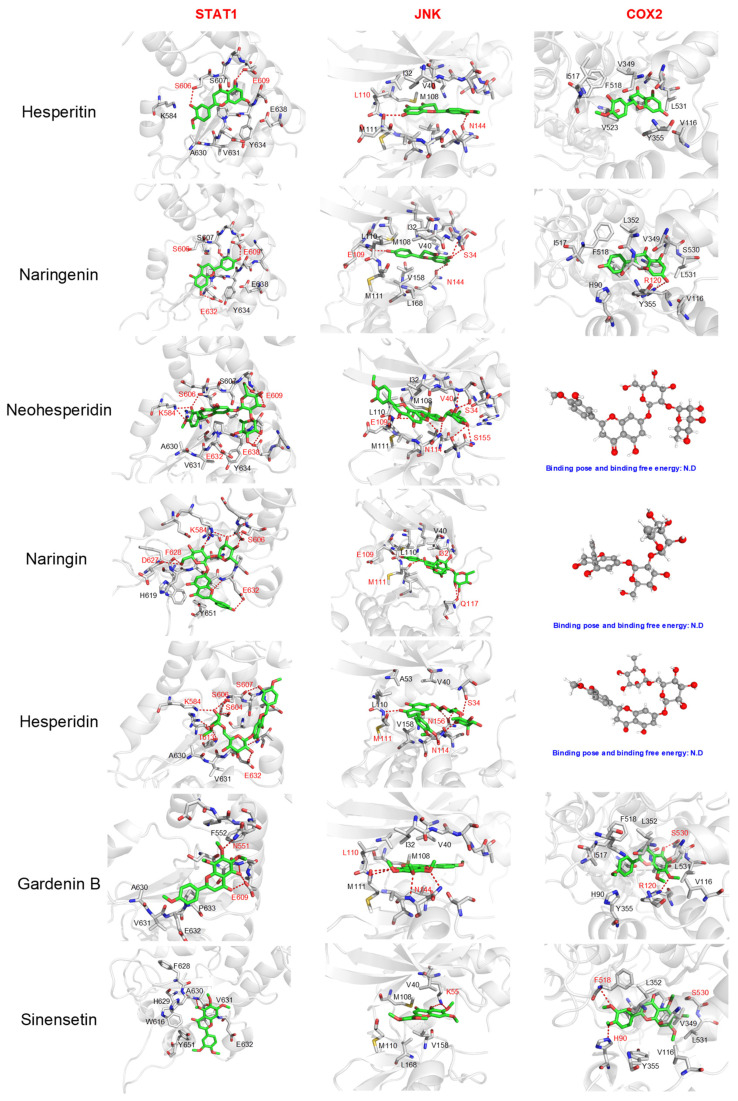
Predicted binding poses of 7 reference compounds towards 3 representative receptors mainly participating in the pro-inflammatory process. The residues involved in polar interactions were colored red, while the residues mainly participating in the non-polar interactions were colored black. The 3D structure of the ligand is shown when the ligand failed to dock into the active site. STAT1: signal transducer and activator of transcription 1; JNK: c-Jun N-terminal kinase 1; COX2: Cyclooxygenase-2.

**Figure 6 molecules-27-05435-f006:**
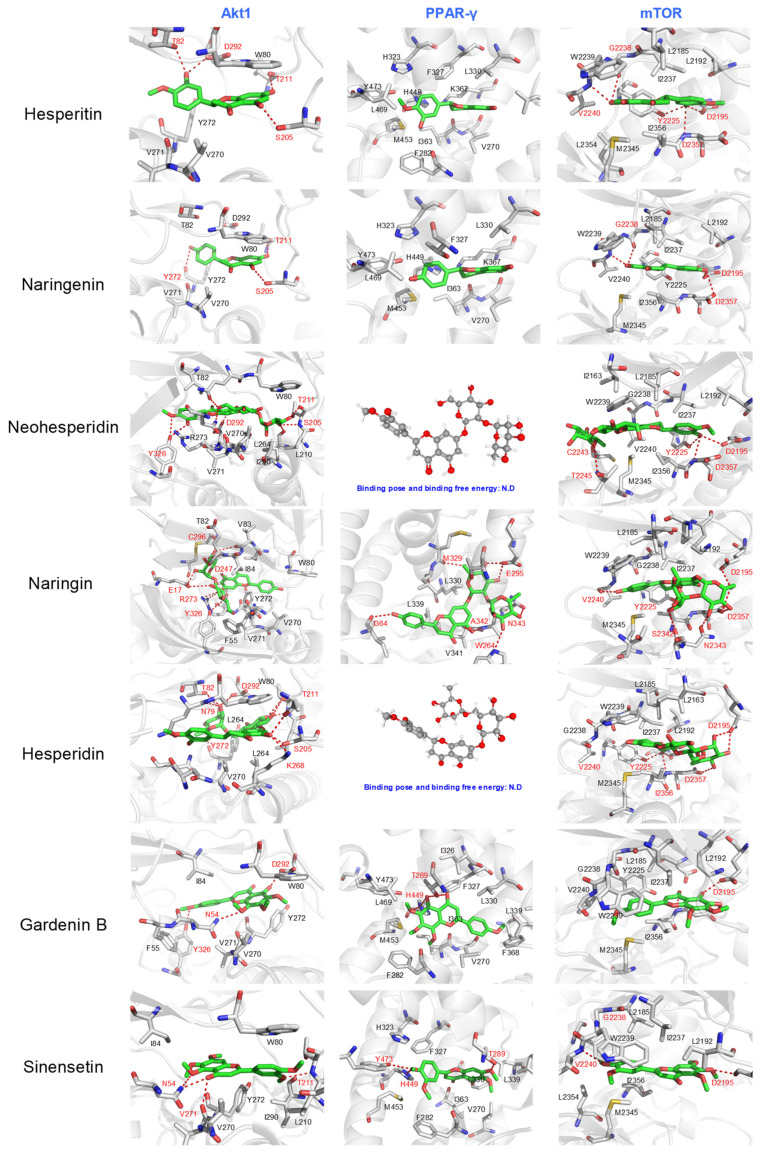
Predicted binding poses of 7 reference compounds towards 3 representative receptors mainly participating in the anti-inflammatory process. The residues involved in polar interactions were colored red, while the residues mainly participating in the non-polar interactions were colored black. The 3D structure of the ligand is shown when the ligand failed to dock into the active site. AKT1: AKT serine/threonine kinase 1; PPAR-γ: peroxisome proliferator activated receptor gamma; mTOR: mechanistic target of rapamycin kinase.

**Figure 7 molecules-27-05435-f007:**
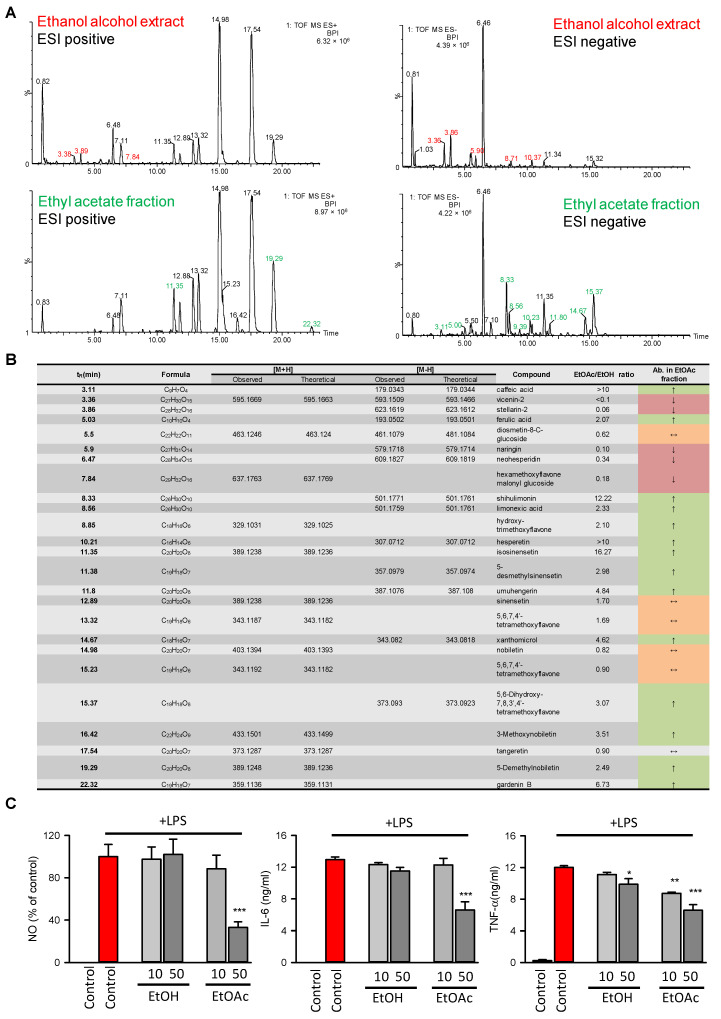
EtOAc extract of CRP showed an increased abundance of phenols and flavonoids, as well as more promising anti-inflammatory potential in vitro than crude EtOH extract. (**A**). HPLC/MS/MS-based chemical profiling demonstrated that phenols and flavonoids were enriched in EtOAc extract. The base peak intensity (BPI) chromatogram of ESI positive and negative ionization mode of crude ethanol extract and EtOAc extract. The retention time was colored red and green to indicate their abundance was decreased or increased in EtOAc extract, respectively. (**B**). The identified compounds in crude EtOH extract and EtOAc extract. The compounds with increased (EtOAC/EtOH ratio greater than 2), decreased (EtOAC/EtOH ratio less than 0.5), or roughly unchanged (EtOAC/EtOH ratio in the range between 0.5 to 2) abundance were noted as “↑”, “↓”, and “↔”, respectively. (**C**). The in vitro anti-inflammatory efficacy of crude EtOH extract and EtOAc extract. The Raw 264.7 cells were treated with crude EtOH extract and EtOAc extract with the indicated concentration in the presence or absence of LPS (100 ng/mL) for 24 h, and 0.1% DMSO was used as the vehicle control. The concentration of NO, IL-6, and TNF-α was measured with assay kits. The pooled data were visualized with mean ± SEM of at least three independent experiments. The one-way ANOVA and Tukey post hoc tests were employed for statistical analysis. *: *p* < 0.05, **: *p* < 0.01, ***: *p* < 0.001.

## Data Availability

Not applicable.

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
