# Peer review of "Bioinformatics-Guided Identification of Ethyl Acetate Extract of Citri Reticulatae Pericarpium as a Functional Food Ingredient with Anti-Inflammatory Potential"

_molecules, 2022, doi:10.3390/molecules27175435_

Round 1
Author Response
Dear Reviewer:
Thank you for the thoughtful comments, which helped us to improve the presentation of our data. We have taken into account all comments and modified the manuscript accordingly.
Please, find below our detailed point by point response in red and changes in text in blue.
Sincerely,
Dr. Lu Jin

Reviewer 2 Report
Starting from the knowledge about Citri Reticulatae Pericarpium (Chenpi), commonly used as food complements and folk medicine, this study aims to investigate especially the anti-inflammatory activity of various compounds extracted from Chenpi. In this regard, in silico approaches (network pharmacology, molecular docking, principal component analysis, and K-means clustering) were used to found potential targets and predict the binding free energy between them and extracted compounds. A comparison between EtOAc extract of Chenpi and crude alcoholic extract showed that the first one might exhibit more promising anti-inflammatory potential, assumption also proved by HPLC/MS/MS-based chemical profiling and in vitro tests. The study is interesting, well presented, detailed and cursive, and the promising results can form the basis for further investigations.
Suggestions:
For a more intuitive approach, I recommend creating a schematic diagram to illustrate the work protocol.
Figure 3C. On what criteria were these compounds chosen to be represented in the poses, considering that, with the exception of Hesperidin, they are not the ones with the lowest binding energy?
Typos: line 72 (anti-oxidative instead of “an-ti-oxidative”; line 236 (delete “naringenin”, the term appears twice).
Author Response

(The authors gave the same response as above.)
